# Advanced stage presentation and its determinant factors among colorectal cancer patients in Amhara regional state Referral Hospitals, Northwest Ethiopia

**Mulugeta Wassie**[ID][1]*, **Debrework Tesgera Beshah**[2], **Yenework Mulu Tiruneh**[2]

**1** Department of Medical Nursing, School of Nursing, College of Medicine and Health Sciences, University of Gondar, Gondar, Ethiopia, **2** Department of Surgical Nursing, School of Nursing, College of Medicine and Health Sciences, University of Gondar, Gondar, Ethiopia

* mulugeta2113@gmail.com

## Abstract

### Introduction

Nowadays, the burden of colorectal cancer (CRC) has been increasing in the world, particularly in developing nations. This could be related to the poor prognosis of the disease due to late presentation at diagnosis and poor treatment outcomes. In Ethiopia, studies related to the stage of colorectal cancer at diagnosis and its determinants are limited. Therefore, the study was intended to assess advanced stage presentation and its associated factors among colorectal cancer patients in northwest Ethiopia.

### Methods

An institution-based retrospective study was conducted among 367 CRC patients at two oncologic centers (the University of Gondar and Felege Hiwot comprehensive specialized hospitals) from January 1, 2017, to December 31, 2020. Data were entered into EPi-data 4.2.0.0 and transferred to STATA version 14 statistical software for analysis. Binary logistic regression was used to identify factors associated with the outcome variable. All variables with P-value < 0.2 during bi-variable analysis were considered for multivariable logistic regression. The level of statistical significance was declared at P-value <0.05.

### Results

The magnitude of advanced stage presentation of colorectal cancer was 83.1%. Being rural dwellers (Adjusted odds ratio (AOR) = 3.6; 95% CI: 1.8,7.2), not medically insured (AOR = 3.9; 95% CI: 1.9,7.8), patients delay (AOR = 6.5; 95% CI:3.2, 13.3), recurrence of the disease (AOR = 2.3; 95% CI: 1.1,4.7), and no comorbidity illness (AOR = 4.4; 95% CI: 2.1, 9.1) were predictors of advanced stage presentation of CRC.

**Data Availability Statement:** All relevant data are within the paper and its Supporting information files.

**Funding:** The author(s) received no specific funding for this work.

**Competing interests:** The authors have declared that no competing interests exist.

**Abbreviations:** AOR, Adjusted Odds Ratio; COR, Crude Odds Ratio; CRC, Colorectal Cancer; FHCSH, Felege Hiwot Comprehensive Specialized Hospital; GUCSH, Gondar University Comprehensive Specialized Hospital.

## Conclusion

The current study revealed that the advanced-stage presentation of colorectal cancer patients was high. It is recommended that the community shall be aware of the signs and symptoms of the disease using different media, giving more emphasis to the rural community, expanding health insurance, and educating patients about the recurrence chance of the disease. Moreover, expansion of colorectal treatment centers and screening of colorectal cancer should be given emphasis.

## Introduction

Colorectal cancer (CRC) is one of the most frequently diagnosed cancers in the world and the second most common oncological cause of death [1]. Its incidence has been gradually increasing, particularly among those who lead a sedentary lifestyle. Moreover, obesity, inactive lifestyle, red meat consumption, alcohol consumption, and tobacco use are considered the pouring factors behind the development of CRC [2].

In all cancer-related deaths among men, 11.2% were owing to CRC and 4.8% of all cancer-related deaths in women were attributed to CRC. Nowadays, it is the most common malignancy representing 13% of all cancers arising in the gastrointestinal tract [2, 3]. In Ethiopia, CRC is the third most prevalent malignancy in the adult populations, and clients often present with late stages of the disease [4]. As to the Addis Ababa cancer registry, CRC is the first in men and fourth in women [5].

Raising awareness of the disease is important to promote healthy lifestyle choices, novel strategies for CRC management, and implementation of global screening programs, which are critical to reducing morbidity and mortality in the future [6]. A significant portion of the CRC survivors had presented with the advanced stage of the disease as reported by different studies conducted around the world [7–10]. Prognosis in CRC is closely related to the disease stage at diagnosis, and patients with early disease may be candidates for curative treatment [7].

Different factors are known to be contributing to the delayed presentation of CRC patients such as; place of residence age of the patients, health insurance, socioeconomic status, screening behavior, health care visit delay, diagnosis delay, shortage of physicians, and accessibility of nearby health facilities [8, 10–15]. Despite studies related to a late-stage presentation of colorectal cancer in different parts of the globe, studies on CRC stage at presentation are limited in Ethiopia, including the study setting. Therefore, the study was aimed at assessing the stage of presentation and its determinant factors among colorectal cancer patients at oncology units in the Amhara region, Northwest Ethiopia.

## Materials and methods

### Study design, period, and settings

An institution-based cross-sectional study was conducted in two oncologic centers in Northwest Amhara regional state of Ethiopia among CRC patients enrolled from January 1, 2017, to December 31, 2020. The data were collected from April 1-May 1, 2021. The two oncologic centers (University of Gondar Comprehensive Specialized Hospital (UOGCSH) and Felege Hiwot Comprehensive Specialized Hospital (FHCSH) are located in the Amhara region, Northwest parts of Ethiopia. UOGCSH is found in Gondar city (one of the ancient cities in Ethiopia), located about 750 km away from Addis Ababa. Currently, UOGCSH is serving more than five million people whereas FHCSH is located in Bahir Dar town, 552 km away from Addis Ababa.

## Study populations

Colorectal cancer patients who had to follow up at UOGCSH and FHCSH from January 1st, 2017 to December 31 2020 were the study populations, and incomplete charts, and charts that were not found during the data collection period were excluded.

## Operational definitions

The stage at diagnosis: The eighth edition of the American Joint Committee of Cancer (AJCC) was used for tumor node metastasis (TNM) staging of CRC [16].

Advanced stage: patients diagnosed with the disease stage III and IV

Early-stage: patients diagnosed with the disease stage I and II

Comorbidity illness: is the occurrence of any disease listed in the Charlson comorbidity index other than colorectal cancer at diagnosis [17].

Carcinoembryonic antigen: Tumor markers of gastrointestinal cancers especially colorectal cancer and classified as elevated ($\geq$ 5ng/ml) and not elevated if <5 ng/ml [18].

Time to visit health facilities (symptom duration): Defined as the duration of time from first perceiving of symptom up to health care visit and it is delayed if a patient seeks health care visit more than ninety days after the symptoms of CRC appeared [7, 19].

Time to diagnosis: the duration of time from initial health care visit up to confirmed diagnosis and delayed if it is >30 days [7, 19].

Diagnosis to treatment interval (DTI). The time interval between confirmed CRC diagnoses up to the date of first treatment started and delayed if it is more than 30 days [7, 19].

## Sample size determination and procedure

All CRC patients who had to follow up at oncology units of two hospitals from January 1, 2017-December 31, 2020 were included to get an adequate sample size since the total number of CRC patients in both oncology units identified from the registration book was 429 (261 at UOGCSH and 168 FHCRH). Out of 429 identified medical record numbers (MRN), 22 charts were mismatched (registered as CRC in the registration book, but different diagnoses in the chart), 23 were not found during the data collection period, and 17 were incomplete. Finally, 367 study participants were included in the study.

## Data collection tools, procedures, and quality

Data were extracted using a pre-tested and structured data abstraction sheet adapted from different kinds of literature [20–26]. The abstraction sheet was developed in English and contains socio-demographic and clinicopathological characteristics of the patients. The data were collected by reviewing medical records. The one-day training was given for data collectors and supervisors on data collection tools and procedures. Two supervisors having master's degrees in clinical oncology nursing were involved. Data collectors were supervised closely by the supervisors and principal investigator. The completeness of each abstraction sheet was checked by the principal investigator and supervisor on a daily base.

## Data processing and analysis

Data were coded, entered, cleaned, and checked by EpiData version 4.6, and analyzed using Stata Version 14.2. Descriptive statistics of different variables were presented using tables, figures, and text. The binary logistic regression model was used to identify independent variables that have associated with the outcome variable (advanced stage presentation). The Hosmer–Lemeshow test was used to test the model fitness of the data and the model fitted with the

insignificant p-value. To reduce potential confounders, bi-variable analysis was employed and independent variables with p-value <0.2 were exported to multivariable analysis. Variables with p -values ≤ 0.05 with 95% confidence interval during multivariable binary logistic regression analysis were declared as statistically significant. In addition, the regression analysis technique was used to eliminate the missed data.

## Ethical considerations

Ethical clearance was obtained from the Ethical Review Committee of the School of Nursing, college of medicine and health sciences, University of Gondar. An official letter was written to GUCSH and FHCSH for permission and support. The research ethics committee of the school of nursing, University of Gondar waived the requirements for informed consent. All data collected from the chart was kept strictly confidential, anonymous, and used only for the study purpose.

## Result

### Socio-demographic characteristics of study participants

A total of 367 study participants were involved. The mean age of the participants was 49.2 years and nearly half, 50.7% were in the age groups greater than50 years. More than half (53.0%) were males and 40.0% were from rural. One hundred forty one (38.4%) had health insurance (Table 1).

### Clinico-pathological characteristics of study participants

Nearly57.0% of study participants presented with an elevated level of carcinoembryonic antigen (CEA), and 22.0% had comorbidity illness. Anemia was diagnosed from nearly 27.3% of the patients. The majority (41.1%) of the patients had colon cancer and 83.0% of them had adenocarcinoma. Regarding the delay, 60.2%, 45.5%, and 29.3% of the participants were delayed for symptoms, diagnosis, and treatment respectively (Table 2).

### Stage of colorectal cancer at the presentation of patients

The current study revealed that advanced stage presentation of colorectal cancer patients was found to be 83.1% (95% confidence intervals (CIs): 78.9%- 86.6%). Nearly two-thirds (62.9%) were diagnosed with stage IV, and only 12 (3.3%) were diagnosed with stage I (Fig 1).

**Table 1. Socio-demographic characteristics of colorectal cancer patients in Northwest Amhara Referral Hospitals (n = 367).**

| Variables | Categories | Frequency | Percentage (%) |
|---|---|---|---|
| Age in years | <50 | 181 | 49.32 |
| | ≥50 | 186 | 50.68 |
| Sex | Male | 193 | 52.59 |
| | Female | 174 | 47.41 |
| Residence | Urban | 148 | 40.33 |
| | Rural | 219 | 59.67 |
| Marital status | Married | 274 | 74.66 |
| | Unmarried | 50 | 13.62 |
| | Widowed | 36 | 9.81 |
| | Divorced | 7 | 1.91 |
| Health insurance | Yes | 141 | 38.42 |
| | No | 226 | 61.58 |

**Table 2. Clinicopathological characteristics of study participants (n = 367).**

| Variables | Categories | Frequency | Percentage (%) |
|---|---|---|---|
| CEA(n = 362) | <5ng/ml | 156 | 43.09 |
| | ≥5ng/ml | 206 | 56.91 |
| Comorbidity illness | Yes | 80 | 21.80 |
| | No | 287 | 78.20 |
| Anemic status | Yes | 100 | 27.25 |
| | No | 267 | 72.75 |
| Tumor location | Ano rectal | 33 | 8.99 |
| | Recto sigmoid | 17 | 4.63 |
| | Colon | 151 | 41.14 |
| | Rectal | 111 | 30.25 |
| | Colorectal | 55 | 14.99 |
| Histological type | Adenocarcinoma | 305 | 83.11 |
| | Mucinous cell carcinoma | 62 | 16.89 |
| CRC recurrence | Yes | 169 | 46.05 |
| | No | 198 | 53.95 |
| Symptom duration | < 3months | 146 | 39.78 |
| | ≥3 months | 221 | 60.22 |
| Diagnosis duration | ≤1 month | 200 | 54.50 |
| | ≥1 month | 167 | 45.50 |
| Diagnosis to treatment | ≤1 month | 244 | 70.72 |
| | ≥1 month | 101 | 29.28 |

## Determinants of advanced-stage presentation of colorectal cancer patients

Both bi-variable and multivariable analysis logistic regression analyses were undertaken. In bi-variable analysis; residence, marital status, insurance status, patient delay, comorbidity illness,

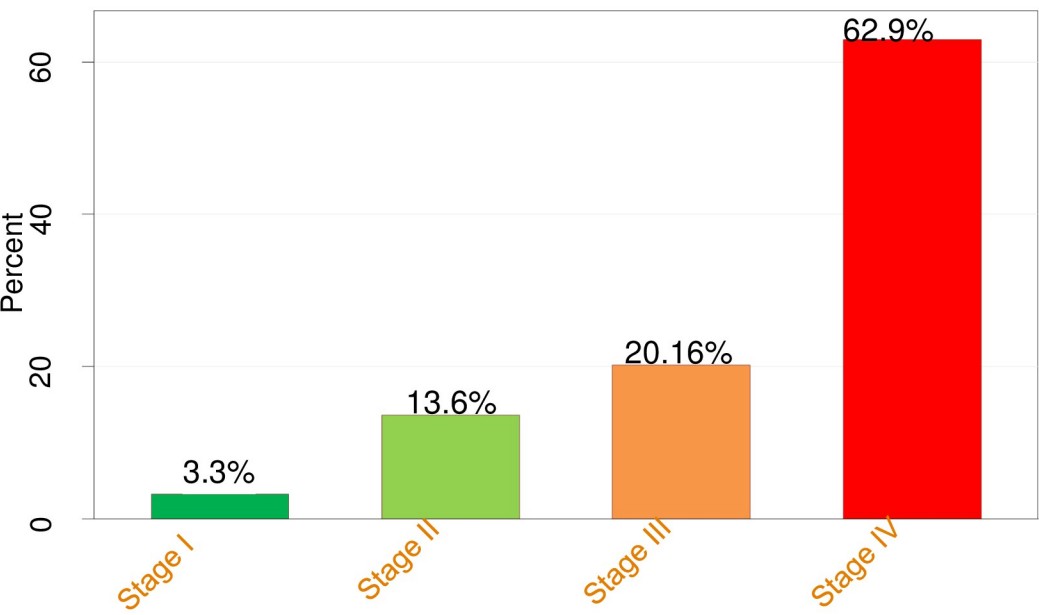

**Fig 1. Distribution of the disease stage on presentation of colorectal cancer patients.**

**Table 3. Determinants of advanced stage presentation among colorectal cancer patients (n = 367).**

| Variables | Category | Advanced stage | | COR(95%CI) | AOR(95%CI) | P-value |
|---|---|---|---|---|---|---|
| | | No (%) | Yes (%) | | | |
| **Residence** | Urban | 42(28.4) | 106(71.6) | | 1 | 1 |
| | Rural | 20(9.1) | 199(90.9) | 3.9 (2.2–7.0)* | 3.6 (1.8–7.2) | <0.001 |
| **Marital status** | Married | 44(16.1) | 230(83.9) | | 1 | 1 |
| | Unmarried | 9(18.0) | 41(82.0) | 0.8(0.4–1.9) | 1.3(0.5–3.5) | 0.539 |
| | Widowed | 5(13.9) | 31(86.1) | 1.2(0.4–3.2) | 2.4(0.6–8.5) | 0.172 |
| | Divorced | 4(57.1) | 3(42.9) | 0.1(0.0–0.6)* | 0.2(0.1–1.2) | 0.079 |
| **Insurance status** | Free | 38(27.0) | 103(73.0) | | 1 | 1 |
| | Paid | 24(10.6) | 202(89.4) | 3.1(1.8–5.5)* | 3.9(1.9–7.8) | <0.001 |
| **Patient delay** | Not delayed | 45(30.2) | 101(69.2) | | 1 | 1 |
| | Delayed | 17(7.7) | 204(92.3) | 5.3(2.9–9.8)* | 6.5(3.2–13.3) | <0.001 |
| **Comorbid illness** | Yes | 27(33.8) | 53(66.0) | | 1 | 1 |
| | No | 35(12.2) | 252(87.8) | 3.6(2.1–6.5)* | 4.4(2.1–9.1) | <0.001 |
| **Disease recurrence** | Yes | 17(10.1) | 152(89.9) | 2.6(1.4–4.8)* | 2.3(1.1–4.7) | 0.023 |
| | No | 45(22.7) | 153(77.3) | | 1 | 1 |
| **Diagnosis delay** | Not delayed | 41(20.5) | 159(79.5) | | 1 | 1 |
| | Delayed | 21(12.6) | 146(87.4) | 1.8(1.0–3.2)* | 1.3(0.6–2.6) | 0.416 |

*P- value <0.2

disease recurrence, and diagnosis delay were fitted at a p-value<0.2. Whereas, in multivariable analysis; place of residency, insurance status, patient delay, comorbidity illness, and disease recurrence was significantly associated with the outcome variable at a p-value<0.05 using 95% CI.

Patients who were from the rural part of the country were 3.6 times more likely to present with advanced stage than those from urban (AOR = 3.6, 95%CI:1.8–7.2). Patients without insurance coverage are 3.9 times more likely to present with advanced stage than non-insurance users (AOR = 3.9; 95% CI: 1.9–7.8). Similarly, patients who delayed health facilities visits were 6.5 times more likely to be presented with advanced stage than patients who were not delayed (AOR = 6.5;95% CI:3.2–13.3). Moreover, patients with no comorbid illness and with recurrence of CRC disease were 4.4 times (AOR = 4.4; 95%CI: 2.1–9.1) and 2.3 times (AOR = 2.3; 95%CI: 1.1–4.7) more likely presented with advanced stage than those with comorbid illness and without disease recurrences, respectively (Table 3).

## Discussion

This study aimed to assess the advanced stage presentation and its determinant factors among CRC patients in Northwest Amhara Referral Hospitals. Place of residency, insurance status, patient delay, comorbid illness, and disease recurrence was significantly associated with the advanced stage of the disease. The study revealed that 83.1% of CRC patients presented at an advanced stage of the disease. This is higher than studies conducted in New York at 60.7% [9], Nottingham at 67% [7], and Stanford at 72% [27]. This could be due to differences in socioeconomic status, accessibility of nearby health facilities, health policies of the respected countries, and patient-health care provider rations of the study areas [28, 29].

Rural dwellers presented with a more advanced stage than those from urban. The finding is supported by different studies [16, 30]. This could be due to inaccessibility of health facilities in the rural community, problems of transportation and probably low awareness of the warning signs of the disease as most rural communities in Ethiopia are illiterate [31].

Patients who hadn't health insurance were presented with advanced stage than those who were insured. This finding is supported by the studies conducted in different countries [8, 11]. The possible justification could be health insurance coverage might solve economic constraints of the patients and visit health care facilities as early as possible if the signs and symptoms of the disease appeared [32, 33].

Patients who delayed visiting a health care facility after the appearance of the first sign/symptom presented with advanced stage than those who did not. This is supported by studies in Puerto Rico [34], Botswana [35], and Spain [36]. This could be since if patients don't visit health care facilities and their physicians as early as possible after the sign/symptom, the disease could metastasize and disseminate to different organs [37].

Patients who hadn't comorbid illness were diagnosed with advanced-stage than those with comorbid illness. This finding is in line with the study conducted in the USA [38], but in contradiction with the literature conducted in different settings [39, 40]. The possible justification could be patients who had comorbidity might visit health care settings more frequently and diagnose with an early stage as another illness [41].

Another contributing factor that caused advanced stage presentation of colorectal cancer patients was a recurrence of the disease. This might be because patients who were diagnosed with colorectal cancer and completed their treatment could not suspect the reoccurrence of the disease despite the presence of signs and symptoms of the disease and causing the delay in their presentation to health care facilities [42]. As such, the government should expand health education regarding CRC symptoms and signs, and options for treatment could increase the early-stage presentation of the disease. As a limitation, the current study didn't consider the economic status of the study participants which could affect the outcome of the study due to the nature of data collection being a chart review.

## Conclusion

The current study revealed that the advanced-stage presentation of colorectal cancer patients was high. Rural residency, no health insurance, delay in health facility visit, absence of comorbidity, and recurrence of the disease significantly increased advanced stage presentation of colorectal cancer. It is recommended that the community shall be aware of the signs and symptoms of the disease using different media, giving more emphasis to the rural community, expanding health insurance, and educating patients about the recurrence rate of the disease. Moreover, expansion of oncology centers, lifestyle modifications, and early screening could be given a great emphasis.

## Supporting information

**S1 Dataset.**
(XLS)

## Acknowledgments

The author's acknowledgments go to UOGCSH and FHCSH medical record room staff, oncology unit staff, and data collectors.

## Author Contributions

**Conceptualization:** Mulugeta Wassie, Debrework Tesgera Beshah, Yenework Mulu Tiruneh.

**Data curation:** Mulugeta Wassie.

**Formal analysis:** Mulugeta Wassie, Debrework Tesgera Beshah, Yenework Mulu Tiruneh.

**Investigation:** Mulugeta Wassie, Debrework Tesgera Beshah, Yenework Mulu Tiruneh.

**Methodology:** Mulugeta Wassie, Yenework Mulu Tiruneh.

**Project administration:** Mulugeta Wassie, Debrework Tesgera Beshah.

**Resources:** Mulugeta Wassie.

**Software:** Mulugeta Wassie, Debrework Tesgera Beshah, Yenework Mulu Tiruneh.

**Supervision:** Mulugeta Wassie.

**Validation:** Mulugeta Wassie.

**Visualization:** Mulugeta Wassie, Yenework Mulu Tiruneh.

**Writing – original draft:** Mulugeta Wassie, Debrework Tesgera Beshah, Yenework Mulu Tiruneh.

**Writing – review & editing:** Mulugeta Wassie, Debrework Tesgera Beshah, Yenework Mulu Tiruneh.

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
