## [Decision Letter · Decision Letter 0]

26 Apr 2022

PONE-D-22-07229Advanced stage presentation and its determinant factors among colorectal cancer patients in Northwest Amhara Regional State Referral Hospitals of EthiopiaPLOS ONE

Dear Dr. Wassie,

Thank you for submitting your manuscript to PLOS ONE. After careful consideration, we feel that it has merit but does not fully meet PLOS ONE’s publication criteria as it currently stands. Therefore, we invite you to submit a revised version of the manuscript that addresses the points raised during the review process.

We look forward to receiving your revised manuscript.

Kind regards,

Negar Rezaei, M.D., Ph.D.,

Academic Editor

PLOS ONE

Journal Requirements:

**Comments to the Author**

1. Is the manuscript technically sound, and do the data support the conclusions?

Reviewer #1: Partly

Reviewer #2: Partly

Reviewer #3: Yes

2. Has the statistical analysis been performed appropriately and rigorously? 

Reviewer #1: No

Reviewer #2: Yes

Reviewer #3: I Don't Know

3. Have the authors made all data underlying the findings in their manuscript fully available?

Reviewer #1: No

Reviewer #2: Yes

Reviewer #3: Yes

4. Is the manuscript presented in an intelligible fashion and written in standard English?

Reviewer #1: No

Reviewer #2: No

Reviewer #3: Yes

5. Review Comments to the Author

Reviewer #1: The authors of this study investigated the advanced stage presentation of colorectal cancer (CRC) and the contributing factors in a regional study in Ethiopia. The study is of high importance since reporting results from a lesser investigated country. Although the manuscript is drafted well some comments and amendments need to be considered prior to any decision on this submission.

1. General: a language and grammar revision are necessary on this submission.

2. Abstract, methods: the definition of “advanced stage CRC” should be provided since it is the primary outcome of the study and its criteria is not clear.

3. Abstract, methods: the precise time span pf study is unclear. Authors mentioned year 2021 and als0 2017-2020 as the investigation period. This point needs a revision.

4. Abstract, results: the variables used to generate the adjusted odds ratio (AOR) should be stated for a clearer data presentation.

5. Keywords: the use of keywords should be based on PubMed MeSH database as much as possible and the authors should choose more relevant ones.

6. Introduction: in third line of the first paragraph, the word “oncological” seems to be misspelled as “ontological”.

7. Introduction: this section needs a major revision to replace the short sentences and paragraphs with 3-4 paragraphs with clear storyline and message. Also, adding some information about the epidemiology of CRC in Ethiopia is essential in this section.

8. Methods, Operational definitions: the primary outcome investigated in this study should be highlighted and also the other variables included in the study and the analyses should be categorized in demographic, clinical, and laboratory sub-classes.

9. Methods, Data processing and analysis: details of the regression analysis are inadequate and the statistical analysis need to be explained clearly in this part.

10. Results: the past tense should be used to report the study results.

11. Results and methods: since the prevalence of the advances stage CRC is relatively high in this study, authors should expand the methods section on the pathologic evaluation of the CRC patients in the two referral hospitals mentioned as the data reference.

12. Results, table 3: the variables included in the two-stages regression analysis need to be expanded to all available variables and factors in this database. For example, demographic variables should be added to the adjustment.

13. Discussion: this section needs one paragraph as the policy implications of the findings of this study to benefit the health authorities in Ethiopia.

Reviewer #2: In the present manuscript, the authors aimed to assess stage of presentation and its contributing factors among colorectal cancer patients in Amhara regional state of Ethiopia. It seems that this study is a part of a project which some of other papers have been previously published. Although it is an important subject, in particular for regional policymakers, there are several issues which should be addressed:

General comments:

1- The use of English needs to be improved within the text. Also, typos and grammatical issues should be addressed in the text and tables.

2- The authors have previously published similar studies on other cancers which can be mentioned or discussed in the manuscript:

https://doi.org/10.1186/s13027-021-00371-6

https://doi.org/10.1186/s12885-019-6447-x

#Introduction

1- The first sentence of Introduction is a basic description which is recommended to be omitted since it is clear to the readers of the journal.

2- There are too many paragraphs in the Introduction, so please merge some of them. This section should be written in almost two to three paragraphs in which epidemiology and risk factors of CRC, determinant factors of late stage and preventive measures for CRC, and gap of knowledge and aim of the study should be explained.

3- Reference 2 would be better to be replaced with a more recent article on the epidemiology of cancers or CRC like the followings:

https://doi.org/10.1016/S2468-1253(19)30345-0

https://doi.org/10.3322/caac.21660

#Methods

1- The inclusion and exclusion criteria should be clarified. For example, were individuals with a specific age range or specific stage of CRC excluded? What about effects of comorbidities on inclusion to the study?

2- Please clarify whether the participants received treatment and if yes, how its effects have been adjusted in the analysis.

3- According to STROBE checklist, please mention statistical methods used to control for confounding and explain how missing data were addressed in the "Data processing and analysis" section.

#Results

1- You might want to use a flow diagram to represent the selection of the study participants.

#Discussion

1- It would be better to mention the study design and participants of similar studies in the Discussion and their findings, then compare the results of the present study with them.

2- If applicable, please cite the relevant studies for your justifications for the differences between the results of the present study with previous studies.

3- It would be great to provide the limitations of the present study in the last paragraph of Discussion.

#Minor comments

1- In the Table 3 descriptions, please define the abbreviations used in the table. Also, please clarify the adjustment was performed for what variables in the table and text.

Reviewer #3: Key Message:

This paper aimed to assess the stage of presentation and its contributing factors among colorectal cancer patients in Amhara regional state of Ethiopia in 2017-2020. As a developing country in Africa, Ethiopia needs more consideration from health policies to decrease the burden of diseases. Cancer diagnosis and management are costly procedures that need proper infrastructure and specialized personnel. Therefore, in countries such as Ethiopia, broadly available infrastructures may be lacking, and considering related studies to investigate the ongoing diseases is critical. Consequently, I think the study carries out important messages about the staging of colorectal cancer patients in the study's region and the associated contributing factors. In overall, the paper was written in an acceptable storyline and presented in a reasonable manner. The methods and results parts were in a better situation, and the introduction and the discussion need revisions in this round.

Evidence and examples:

Major issues:

The discussion was mainly a narration and simply a review of the results. The associations of some factors seem to be controversial. In these situations citing enough and proper evidence is highly recommended. Therefore, despite the storyline of the discussion is currently acceptable, it is recommended revising the discussion comprehensively.

In the conclusion part it was just stated: "The current study revealed that advanced stage presentation of colorectal cancer patients was high". But as we can see, the aim of the study was "to assess stage of presentation and its contributing factors among colorectal cancer patients in Amhara regional state of Ethiopia". As we can see, the conclusion lacks findings of contributing factors.

The introduction consisted of some tiny paragraphs. Combining related paragraphs as a whole is highly recommended. The storyline of the introduction should be considered carefully. The introduction may be started from what currently we know (known of evidence) in the first paragraphs, and continued by what we don't know (gap of knowledge) in the next paragraph, and finally completed by what we have done to fill the gap of knowledge and the aim of the study in the last paragraph.

The draft had some grammatical, English fluency and readability issues. Please address these issues by the assistance of an English expert.

Minor issues:

In the introduction in the sentence "CRC incidence has been gradually increasing in the globe, particularly in developing nations who lead sedentary lifestyle." I cannot understand what the authors mean? If you mean developing countries are the leading countries in sedentary lifestyle, I do not agree and if you disagree with me, it is suggested to prove this hypothesis by citing proper evidence.

In the introduction in the sentence "Obesity, inactive lifestyle, red meat consumption, alcohol consumption, and tobacco are considered the pouring factors behind the development of CRC", the "pouring factors" is a vague term and I am not familiar with it.

As this study was partly a descriptive study, it is recommended to add the study period to the aim of the study.

In the methods in the sentence "Carcino embryonic antigen: Tumor markers of gastric cancer especially colorectal cancer and classified as elevated (≥ 5ng/ml) and not elevated if <5 ng/ml", the "gastric cancer" should be revised. Because in medicine, gastric cancer is related to the stomach, and I think here the authors mean gastrointestinal cancers.

In the methods in the sentence "Time to seek health care facility (symptom duration):" as we can see the definition, instead of "time to seek" it is "time to visit" or "time to receive health service" because in the Health Care Access science it is defined that from seeking the healthcare to receiving that also takes considerable time, especially in resource-limited communities.

It is highly recommended to attach the structured data abstraction sheet to the appendix of the paper.

In the methods, please mention the level of education and experience of supervisors.

In the results, it is recommended to report all percentages with one decimal.

At the beginning of the discussion, main findings of the study should be presented. In the current draft, we cannot see enough information for the first paragraph of the discussion. As the authors explained in the introduction, this study was aimed to "assess stage of presentation and its contributing factors among colorectal cancer patients in Amhara regional state of Ethiopia". First, restating the aim of the study in the discussion is not necessary. Second, in the first paragraph of the discussion, the readers are willing to read about which factors are related to the advanced stage of CRC.

In the second paragraph of the discussion starting paragraph with "this" is not suitable. And this paragraph seems to be a continuation of the first paragraph and can be merged into that.

In the discussion, the paragraph started with "Patients who hadn't comorbidity illness were diagnosed with advanced stage than those who had" can be controversial findings. Therefore, it is highly recommended to discuss this issue carefully and cite more studies.

In the discussion in the sentence "Another contributing factor that caused delayed presentation of colorectal cancer patients was recurrence of the disease". The "delayed presentation" is vague and can be assumed as "advanced stage presentation" or "delay of receiving treatment".

6. PLOS authors have the option to publish the peer review history of their article (what does this mean?). If published, this will include your full peer review and any attached files.

Reviewer #1: **Yes: **Sina Azadnajafabad, MD, MPH

Reviewer #2: No

Reviewer #3: **Yes: **Mohammad-Mahdi Rashidi

---

## [Author Response · Author response to Decision Letter 0]

1 Jun 2022

01/06/2022

PLOS ONE

Subject: Submission of revised manuscript entitled as “Advanced stage presentation and its determinant factors among colorectal cancer patients in Northwest Amhara Regional State Referral Hospitals of Ethiopia (PONE-D-22-07229).

Dear,Negar Rezaei M.D., Ph.D, thank you for your email enclosing the Editorial member’s and the reviewer’s comments. We have carefully revised the manuscript and incorporated their comments accordingly. Our responses are given in point-by-point response below.

We hope the revised version is suitable for publication and look forward to hearing from you in due courses.

Sincerely

Mulugeta Wassie. 

University of Gondar, College of Medicine and health Sciences, School of Nursing, Department of Medical Nursing. 

Point by point responses to Editorial Board Member’s and Reviewers’ comments

Academic Editor

Authors’ response: Thank you very much for your constructive suggestions and comments. We have tried to abide with the PLOS ONE style requirements

Authors’ response: Thank you very much for your constructive suggestions. We have used a retrospective study of medical records and we have discussed it in ethics statement of the manuscript as all data were fully anonymized. The research ethics committee of school of nursing, University of Gondar waived the requirements for informed consent.

Authors’ response: Thank you very much for your suggestions. We have updated the data availability statement and we uploaded the minimal data used to extract the report in the form of excel in “supplementary files” section.

Reviewer #1:

 The authors of this study investigated the advanced stage presentation of colorectal cancer (CRC) and the contributing factors in a regional study in Ethiopia. The study is of high importance since reporting results from a lesser investigated country. Although the manuscript is drafted well some comments and amendments need to be considered prior to any decision on this submission.

1. General: a language and grammar revision is necessary on this submission.

Authors’ response: Thank you very much for your review and constructive comments provided. We have tried to revise the whole manuscript to address the language and grammatical errors.

2. Abstract, methods: the definition of “advanced stage CRC” should be provided since it is the primary outcome of the study and its criteria is not clear.

Authors’ response: Thank you very much for your constructive comments. The authors have tried to address the comments in the manuscript.

3. Abstract, methods: the precise time span of study is unclear. Authors mentioned year 2021 and also 2017-2020 as the investigation period. This point needs a revision.

Authors’ response: Thank you very much for your constructive comments. Sorry for the unclarity. “April 1-May 1, 2021” was the actual data collection period and “2017-2020” was the period of patients’ registration(admission) in the each respective hospitals as this study is retrospective chart review study. We also tried to clear it in the abstract section of the document.

4. Abstract, results: the variables used to generate the adjusted odds ratio (AOR) should be stated for a clearer data presentation.

Authors’ response: Thank you very much for your constructive comments. We have stated the variables used to generate AOR in the abstract section as per the comment.

5. Keywords: the use of keywords should be based on PubMed MeSH database as much as possible and the authors should choose more relevant ones.

Authors’ response: Thank you very much for your constructive comments. We have tried to use MeSH database to select the Keywords and modified in the manuscript accordingly.

6. Introduction: in third line of the first paragraph, the word “oncological” seems to be misspelled as “ontological”.

Authors’ response: Thank you very much for your critical review. We have corrected it in the main document

7. Introduction: this section needs a major revision to replace the short sentences and paragraphs with 3-4 paragraphs with clear storyline and message. Also, adding some information about the epidemiology of CRC in Ethiopia is essential in this section.

Authors’ response: Thank you very much for your constructive comments. We have tried to address the comments and suggestions in the manuscript.

8. Methods, Operational definitions: the primary outcome investigated in this study should be highlighted and also the other variables included in the study and the analyses should be categorized in demographic, clinical, and laboratory sub-classes.

Authors’ response: Thank you very much for your constructive comments. We have tried to address the comments and suggestions in the manuscript.

9. Methods, Data processing and analysis: details of the regression analysis are inadequate and the statistical analysis needs to be explained clearly in this part.

Authors’ response: Thank you very much for your comments. The authors have tried to explain more about the regression analysis in the revised manuscript. 

10. Results: the past tense should be used to report the study results.

Authors’ response: Thank you very much for your comments. We have rewritten in the form of past tense

11. Results and methods: since the prevalence of the advances stage CRC is relatively high in this study, authors should expand the methods section on the pathologic evaluation of the CRC patients in the two referral hospitals mentioned as the data reference.

Authors’ response: Thank you very much for your comments. The authors have tried to assess the pathological evaluation of CRC and presented it in the result section like histological type of disease, CEA level …..

12. Results, table 3: the variables included in the two-stage regression analysis need to be expanded to all available variables and factors in this database. For example, demographic variables should be added to the adjustment.

Authors’ response: Thank you very much for your comments. Firs, the authors have employed bivariable (all independent variables individually regressed against the dependent variable) regression analysis to identify the potential confounders. And those variables with p-value <0.2 were identified and included in the two stage regression (multivariable analysis) and variables with a p-value<0.05 were declared as significantly affected the outcome variable. It shows that all available variables including socio demographic variables were included step by step (from bi variable to multi variable) analysis.

13. Discussion: this section needs one paragraph as the policy implications of the findings of this study to benefit the health authorities in Ethiopia.

Authors’ response: Thank you very much for your comments. We have added one paragraph on the implication of the findings with recommendations

Reviewer #2: 

In the present manuscript, the authors aimed to assess stage of presentation and its contributing factors among colorectal cancer patients in Amhara regional state of Ethiopia. It seems that this study is a part of a project which some of other papers have been previously published. Although it is an important subject, in particular for regional policymakers, there are several issues which should be addressed:

General comments:

1- The use of English needs to be improved within the text. Also, typos and grammatical issues should be addressed in the text and tables.

Authors’ response: Thank you very much for your comments. We have tried to improve the grammar and typos of the manuscript in the main document.

2- The authors have previously published similar studies on other cancers which can be mentioned or discussed in the manuscript:

https://doi.org/10.1186/s13027-021-00371-6, https://doi.org/10.1186/s12885-019-6447-x

Authors’ response: Thank you very much for your comments. As you have mentioned, the publications (https://doi.org/10.1186/s13027-021-00371-6, https://doi.org/10.1186/s12885-019-6447-x ) are the publications of the primary author of this manuscript. However, these articles are conducted among CERVICAL CANCER patients who haven’t any similarity of the current study. Due to dissimilarity of the study population and the malignancy type, the authors intentionally left to discuss these articles with the current manuscript. Also there is no space to discuss or include such articles in the current document.

#Introduction

1- The first sentence of Introduction is a basic description which is recommended to be omitted since it is clear to the readers of the journal.

Authors’ response: Thank you very much for your comments. We have omitted the first sentence of the introduction.

2- There are too many paragraphs in the Introduction, so please merge some of them. This section should be written in almost two to three paragraphs in which epidemiology and risk factors of CRC, determinant factors of late stage and preventive measures for CRC, and gap of knowledge and aim of the study should be explained.

Authors’ response: Thank you very much for your suggestion and comments. We have merged those fragmented paragraphs and added some other important information in the main document.

3- Reference 2 would be better to be replaced with a more recent article on the epidemiology of cancers or CRC like the followings:

https://doi.org/10.1016/S2468-1253(19)30345-0, https://doi.org/10.3322/caac.21660

Authors’ response: Thank you very much for cooperation and suggestion. We have incorporated the recent articles with their references as per your suggestion in the main document.

#Methods

1- The inclusion and exclusion criteria should be clarified. For example, were individuals with a specific age range or specific stage of CRC excluded? What about effects of comorbidities on inclusion to the study?

Authors’ response: Thank you very much for your comments. Patients with age groups less than 15 were excluded in the current study since patients with this age group are treated in pediatric oncology ward and the occurrence of CRC in this age group is very rare. It has been included in the revised manuscript. But there was no any exclusion for patients with specific stage or with comorbidity. Comorbidity was one of the “independent variables” and there is no any reason to exclude those patients with comorbidity. If we were studying mortality of CRC, patients with comorbid illness could be excluded. But, currently we are studying about STAGE. We think, treating comorbidity as an explanatory variable weather it affected early or late presentation of the patients is enough.

2- Please clarify whether the participants received treatment and if yes, how its effects have been adjusted in the analysis.

Authors’ response: Thank you very much for your concern. We think treatment has no any effect for the current study as it is about the BASELINE PRESENTATION STAGE. Furthermore, since we are investigating baseline presentation, all patients did not get any treatment and it is default.

3- According to STROBE checklist, please mention statistical methods used to control for confounding and explain how missing data were addressed in the "Data processing and analysis" section.

Authors’ response: Thank you very much for your comments and suggestions.

The Hosmer–Lemeshow test was used to test the model fitness of the data and binary logistic regression model has fitted with insignificant p-value (p-value >0.05). Bi-variable analysis was used to reduce confounding variables. Variables with p-value less than 0.2 in bivariable analysis were included in multivariable analysis.

Regarding the treating of missing data, regression analysis technique was used to systematically eliminate the missed data. Regression analysis technique uses the lowest sample size in the subgroup analysis. In fact there was insignificant missing value in our dataset. It appears only in one variable subgroup (CEA with n=362) out of the total (376).

#Results

1- You might want to use a flow diagram to represent the selection of the study participants.

Authors’ response: Thank you very much for your suggestion. The authors believe that it is not appropriate to put the flow diagram to represent the selection of the study participants in the RESULT section. If it is much important to use a flow diagram, we can put it in the methods and material section of the main manuscript.

#Discussion

1- It would be better to mention the study design and participants of similar studies in the Discussion and their findings, then compare the results of the present study with them.

Authors’ response: Thank you very much for your suggestions and comments. It has been addressed as per the comments in the main document.

2- If applicable, please cite the relevant studies for your justifications for the differences between the results of the present study with previous studies.

Authors’ response: Thank you very much for your suggestions and comments. We have tried to cite the relevant studies to strengthen the justifications of similarities and dissimilarities with the current study.

3- It would be great to provide the limitations of the present study in the last paragraph of Discussion.

Authors’ response: Thank you very much for your suggestions and comments. We have provided the limitations of the current study in the specified space.

#Minor comments

1- In the Table 3 descriptions, please define the abbreviations used in the table. Also, please clarify the adjustment was performed for what variables in the table and text.

Authors’ response: Thank you very much for your suggestions and comments. We have tried to address the comments as per the suggestions and the comments.

Reviewer #3: 

Key Message: 

This paper aimed to assess the stage of presentation and its contributing factors among colorectal cancer patients in Amhara regional state of Ethiopia in 2017-2020. As a developing country in Africa, Ethiopia needs more consideration from health policies to decrease the burden of diseases. Cancer diagnosis and management are costly procedures that need proper infrastructure and specialized personnel. Therefore, in countries such as Ethiopia, broadly available infrastructures may be lacking, and considering related studies to investigate the ongoing diseases is critical. Consequently, I think the study carries out important messages about the staging of colorectal cancer patients in the study's region and the associated contributing factors. In overall, the paper was written in an acceptable storyline and presented in a reasonable manner. The methods and results parts were in a better situation, and the introduction and the discussion need revisions in this round.

Evidence and examples:

Major issues:

The discussion was mainly a narration and simply a review of the results. The associations of some factors seem to be controversial. In these situations citing enough and proper evidence is highly recommended. Therefore, despite the storyline of the discussion is currently acceptable, it is recommended revising the discussion comprehensively.

Authors’ response: Thank you very much for your suggestions and comments. The authors have tried to revise the discussion section of the manuscript as per the comments.

****

In the conclusion part it was just stated: "The current study revealed that advanced stage presentation of colorectal cancer patients was high". But as we can see, the aim of the study was "to assess stage of presentation and its contributing factors among colorectal cancer patients in Amhara regional state of Ethiopia". As we can see, the conclusion lacks findings of contributing factors.

Authors’ response: Thank you very much for your comments. We have included the contributing factors in the conclusion of the revised manuscript.

****

The introduction consisted of some tiny paragraphs. Combining related paragraphs as a whole is highly recommended. The storyline of the introduction should be considered carefully. The introduction may be started from what currently we know (known of evidence) in the first paragraphs, and continued by what we don't know (gap of knowledge) in the next paragraph, and finally completed by what we have done to fill the gap of knowledge and the aim of the study in the last paragraph.

Authors’ response: Thank you very much for your suggestions and comments. We have revised the introduction section of the revised manuscript rigorously.

****

The draft had some grammatical, English fluency and readability issues. Please address these issues by the assistance of an English expert.

Authors’ response: Thank you very much for your suggestions and comments. We have tried to address the raised issues in the revised manuscript.

****

Minor issues: 

In the introduction in the sentence "CRC incidence has been gradually increasing in the globe, particularly in developing nations who lead sedentary lifestyle." I cannot understand what the authors mean? If you mean developing countries are the leading countries in sedentary lifestyle, I do not agree and if you disagree with me, it is suggested to prove this hypothesis by citing proper evidence.

Authors’ response: Thank you very much for your suggestions. In this sentence the authors have wanted to pass the information “those individuals leading sedentary life are more prone to develop CRC even they are living in developing nations as compared to those who don’t have sedentary life style”. Nowadays, sedentary life style (using vehicles even for very short distance, long time sitting in the office, absence of regular physical activities….) in developing nations becomes more common than before. But it is not mean that developing nations lead more sedentary lifestyle than developed nations. We have tried to modify the sentence in the revised manuscript.

****

In the introduction in the sentence "Obesity, inactive lifestyle, red meat consumption, alcohol consumption, and tobacco are considered the pouring factors behind the development of CRC", the "pouring factors" is a vague term and I am not familiar with it.

Authors’ response: Thank you very much for your concern. Sorry for the inconvenience. “Pouring factors” in this sentence is to mean “the driving factors”.

****

As this study was partly a descriptive study, it is recommended to add the study period to the aim of the study.

Authors’ response: Thank you very much for your suggestions. We have added the study period in the revised manuscript

****

In the methods in the sentence "Carcino embryonic antigen: Tumor markers of gastric cancer especially colorectal cancer and classified as elevated (≥ 5ng/ml) and not elevated if <5 ng/ml", the "gastric cancer" should be revised. Because in medicine, gastric cancer is related to the stomach, and I think here the authors mean gastrointestinal cancers.

Authors’ response: Thank you very much for your critical review and comments. We have changed the word “gastric cancer” to “gastrointestinal cancers” in the revised manuscript.

****

In the methods in the sentence "Time to seek health care facility (symptom duration):" as we can see the definition, instead of "time to seek" it is "time to visit" or "time to receive health service" because in the Health Care Access science it is defined that from seeking the healthcare to receiving that also takes considerable time, especially in resource-limited communities. It is highly recommended to attach the structured data abstraction sheet to the appendix of the paper.

Authors’ response: Thank you very much for your comment. We really appreciate your critical and professional review of the document. As per the comment we have changed “Time to seek health care facility” by "time to visit health facilities".

Regarding the “data abstract sheet”, the journal manuscript preparation style doesn’t allow attaching it as an appendix and there is no space to attach the abstract sheet in the manuscript tracking system of the journal. Rather, the journal needs ‘the dataset’ in which the report is extracted from and we have attached the dataset as the supplementary file.

****

In the methods, please mention the level of education and experience of supervisors.

Authors’ response: Thank you very much for your suggestion. We have mentioned the level of education and experience of supervisors.

****

In the results, it is recommended to report all percentages with one decimal.

Authors’ response: Thank you very much for your suggestion. We have tried to revise the result section as per the comment.

****

At the beginning of the discussion, main findings of the study should be presented. In the current draft, we cannot see enough information for the first paragraph of the discussion. As the authors explained in the introduction, this study was aimed to "assess stage of presentation and its contributing factors among colorectal cancer patients in Amhara regional state of Ethiopia". First, restating the aim of the study in the discussion is not necessary. Second, in the first paragraph of the discussion, the readers are willing to read about which factors are related to the advanced stage of CRC.

Authors’ response: Thank you very much for your comments. Dear reviewer, some scholars agree with your idea about the introduction paragraph of the discussion. On the other side, there are also some scholars strongly recommend to introduce the aim of the study in the first section of the discussion. If it violates scholarly writing style, we can remove it, but now we leave as it is. Regarding the contributing factors, we have added variables that affected the outcome variable in the recommended place.

****

In the second paragraph of the discussion starting paragraph with "this" is not suitable. And this paragraph seems to be a continuation of the first paragraph and can be merged into that.

Authors’ response: Thank you very much for your comments. We have merged it as per the comment.

****

In the discussion, the paragraph started with "Patients who hadn't comorbidity illness were diagnosed with advanced stage than those who had" can be controversial findings. Therefore, it is highly recommended to discuss this issue carefully and cite more studies.

Authors’ response: Thank you very much for your comments: As you mentioned, it seems controversial. But the reality seems true. Patients with comorbidity might have frequent contact with health care professionals for the reason other than CRC. This frequent contact could help the patients and health care professionals to detect the early signs and symptoms of the CRC. Therefore, this might help the patients with comorbid illness to be diagnosed at the early stage of the disease than those who haven’t. This is also supported by different literatures (Zafar SY, Abernethy AP, Abbott DH, Grambow SC, Marcello JE, Herndon JE 2nd, Rowe KL, Kolimaga JT, Zullig LL, Patwardhan MB, Provenzale DT. Comorbidity, age, race and stage at diagnosis in colorectal cancer: a retrospective, parallel analysis of two health systems. BMC Cancer. 2008 Nov 25;8:345. doi: 10.1186/1471-2407-8-345. PMID: 19032772; PMCID: PMC2613913.)

****

In the discussion in the sentence "Another contributing factor that caused delayed presentation of colorectal cancer patients was recurrence of the disease". The "delayed presentation" is vague and can be assumed as "advanced stage presentation" or "delay of receiving treatment"

Authors’ response: Thank you very much for your comments. As per your comment, we have changed the "delayed presentation" to "advanced stage presentation".

---

## [Decision Letter · Decision Letter 1]

17 Jun 2022

PONE-D-22-07229R1Advanced stage presentation and its determinant factors among colorectal cancer patients in Northwest Amhara Regional State Referral Hospitals of EthiopiaPLOS ONE

Dear Dr. Wassie,

Thank you for submitting your manuscript to PLOS ONE. After careful consideration, we feel that it has merit but does not fully meet PLOS ONE’s publication criteria as it currently stands. Therefore, we invite you to submit a revised version of the manuscript that addresses the points raised during the review process.

It is strongly recommended general language editing and revisions regarding grammar and syntax by a native speaker or an expert since there are several grammatical errors and typos in the text.

We look forward to receiving your revised manuscript.

Kind regards,

Negar Rezaei, M.D., Ph.D.,

Academic Editor

PLOS ONE

Journal Requirements:

Additional Editor Comments (if provided):

There are two issues regarding this manuscript.

1. There is a questionable difference between the manuscript-mentioned study period and the dates available in the datasheet

2. A native language edit of manuscript is recommended.

Reviewers' comments:

Reviewer's Responses to Questions

**Comments to the Author**

1. If the authors have adequately addressed your comments raised in a previous round of review and you feel that this manuscript is now acceptable for publication, you may indicate that here to bypass the “Comments to the Author” section, enter your conflict of interest statement in the “Confidential to Editor” section, and submit your "Accept" recommendation.

Reviewer #1: (No Response)

Reviewer #2: (No Response)

Reviewer #3: (No Response)

2. Is the manuscript technically sound, and do the data support the conclusions?

Reviewer #1: Yes

Reviewer #2: Yes

Reviewer #3: Yes

3. Has the statistical analysis been performed appropriately and rigorously? 

Reviewer #1: Yes

Reviewer #2: Yes

Reviewer #3: Yes

4. Have the authors made all data underlying the findings in their manuscript fully available?

Reviewer #1: Yes

Reviewer #2: No

Reviewer #3: Yes

5. Is the manuscript presented in an intelligible fashion and written in standard English?

Reviewer #1: No

Reviewer #2: No

Reviewer #3: Yes

6. Review Comments to the Author

Reviewer #1: The authors tried to address my comments and suggestions in this revision and I can say they were successful to some extent. Most of the scientific comments have been addressed. One major issue still remains that is the several grammatical and typo errors and the poor language of this manuscript. The prepared draft needs a revision by a native editor or English expert and the language and structure of manuscript should be revised in this regard, otherwise it may jeopardize the efforts of the authors.

Reviewer #2: Thank you for implementing the comments. Manuscript will require thorough proofreading to meet the required standards of language.

Reviewer #3: In this manuscript, the authors aimed to assess stage of presentation and its contributing factors among colorectal cancer patients in Amhara regional state of Ethiopia. The manuscript was reviewed once in the previous round and revised by the authors. I reviewed previous comments and authors' responses, and most of them were adequate. Just to mention some English fluency issues still were in the manuscript. A copyediting by an expert is recommended.

As the academic editor requested uploading the minimal anonymized dataset, the authors uploaded the minimal data used to extract the report in the form of a datasheet in the supplements. As I reviewed the datasheet two issues raised some questions. First, the number of participants was 367 in the datasheet, which was also mentioned in the last sentence of the "Sample size determination and procedure" subheading in "Materials and Methods", but as we can see in all other parts of the manuscript the total number of participants was reported 376.

Another issue is that in the "Study design, Period, and settings" part, it was mentioned that "patients enrolled from January 1, 2017, to December 31, 2020", but in the datasheet, "Date_of_last_visit" is from 2009 to 2013. As it seems from this evidence the participants' recruitments were not from 2017 to 2020. I would like to receive the authors' response in this regard.

7. PLOS authors have the option to publish the peer review history of their article (what does this mean?). If published, this will include your full peer review and any attached files.

Reviewer #1: **Yes: **Sina Azadnajafabad, MD, MPH

Reviewer #2: No

Reviewer #3: No

---

## [Author Response · Author response to Decision Letter 1]

23 Jul 2022

Manuscript ID: PONE-D-22-07229R1

Topic: Advanced stage presentation and its determinant factors among colorectal cancer patients in Northwest Amhara Regional State Referral Hospitals of Ethiopia

Dear Editors and reviewers;

We sincerely appreciate the valuable comments and suggestions from the reviewers and editors side. The suggestions and comments have been closely followed and revisions have been made accordingly. The following are the questions extracted from the reviewers’ comments along with our summarized responses. Thank you very much for your constructive comments. We tried to inculcate your comments and questions as described below. The changes are attached with. Much correction has been made through the document.

Editor comments

Editor’s comment #01: There is a questionable difference between the manuscript-mentioned study period and the dates available in the datasheet.

Author’s response: 

Thank you very much for your constructive comments: Your concern is quite right. Sorry for the inconvenience. But there is a difference in Ethiopian calendar (E.C) and Gregorian calendar (G.C). On the dataset, time of study participants enrollment was by Ethiopian calendar since the data is registered in E.C. But the time in the manuscript was written by G.C. By adding 8 years (from January to August) on 2009 E.C equals with 2017 in G.C and adding 7 years (from September to December) on 2013 E.C equals 2020 G.C. This 8 and 7 years difference is due to the new year of January and September in G.C and E.C respectively. During analysis we changed Ethiopian calendar to Gregorian calendar. Kindly understand it in such a manner.

Editor’s comment #02: A native language edit of manuscript is recommended.

Author’s response: Thank you very much for the comment: sure! The manuscript has been reviewed by different personnel and many editorial and grammatical issues have been addressed. 

Reviewer #01

Reviewer 1 comments and suggestion #01: The authors tried to address my comments and suggestions in this revision and I can say they were successful to some extent. Most of the scientific comments have been addressed. One major issue still remains that is the several grammatical and typo errors and the poor language of this manuscript. The prepared draft needs a revision by a native editor or English expert and the language and structure of manuscript should be revised in this regard, otherwise it may jeopardize the efforts of the authors.

Author’s response:

 Thank you very much for your critical comments for the improvement of the manuscript. The manuscript has been reviewed by different personnel and many editorial and grammatical issues have been addressed.

Reviewer #02

Reviewer 2 comments and suggestion #01: Thank you for implementing the comments. Manuscript will require thorough proofreading to meet the required standards of language.

Author’s response: Thank you very much for your critical comments for the improvement of the manuscript. The manuscript has been reviewed by different personnel and many editorial and grammatical issues have been addressed.

Reviewer #03

Reviewer 3 comments and suggestion #01: In this manuscript, the authors aimed to assess stage of presentation and its contributing factors among colorectal cancer patients in Amhara regional state of Ethiopia. The manuscript was reviewed once in the previous round and revised by the authors. I reviewed previous comments and authors' responses, and most of them were adequate. Just to mention some English fluency issues still were in the manuscript. A copyediting by an expert is recommended. As the academic editor requested uploading the minimal anonym zed dataset, the authors uploaded the minimal data used to extract the report in the form of a datasheet in the supplements. As I reviewed the datasheet two issues raised some questions. First, the number of participants was 367 in the datasheet, which was also mentioned in the last sentence of the "Sample size determination and procedure" subheading in "Materials and Methods", but as we can see in all other parts of the manuscript the total number of participants was reported 376. Another issue is that in the "Study design, Period, and settings" part, it was mentioned that "patients enrolled from January 1, 2017, to December 31, 2020", but in the datasheet, "Date_of_last_visit" is from 2009 to 2013. As it seems from this evidence the participants' recruitments were not from 2017 to 2020. I would like to receive the authors' response in this regard.

Author’s response:

 Thank you very much for your critical comments. The manuscript has been reviewed by different personnel and many editorial and grammatical issues have been addressed. You are right about the sample size difference but this is due to type errors, the exact number of sample size is 367. 

The other concern that you raise is about enrollments of study participants: 

Your concern is quite right. Sorry for the inconvenience. But there is a difference in Ethiopian calendar (E.C) and Gregorian calendar (G.C). On the datasheet time of study participants enrollment was by Ethiopian calendar since the data is registered in E.C. But the time in the manuscript was written by G.C. By adding 8 years (from January to August) on 2009 E.C equals with 2017 in G.C and adding 7 years (from September to December) on 2013 E.C equals 2020 G.C. This 8 and 7 years difference is due to the new year of January and September in G.C and E.C respectively. During analysis we changed Ethiopian calendar to Gregorian calendar. Kindly understand it in such a manner.

---

## [Decision Letter · Decision Letter 2]

4 Aug 2022

PONE-D-22-07229R2Advanced stage presentation and its determinant factors among colorectal cancer patients in Amhara Regional State Referral Hospitals, Northwest EthiopiaPLOS ONE

Dear Dr. Wassie,

Thank you for submitting your manuscript to PLOS ONE. After careful consideration, we feel that it has merit but does not fully meet PLOS ONE’s publication criteria as it currently stands. Therefore, we invite you to submit a revised version of the manuscript that addresses the points raised during the review process.

We look forward to receiving your revised manuscript.

Kind regards,

Negar Rezaei, M.D., Ph.D.,

Academic Editor

PLOS ONE

Journal Requirements:

Additional Editor Comments (if provided):

Please consider the reviewer 3 comments as follow:

Some copy editing and other issues are still in the draft.

Please add the confidence interval (CI) measurement definition to the analysis section of the methods.

Some percentages were presented without decimals, some with one decimal, and some with two decimals. To maintain consistency, I suggest presenting all the percentages with one decimal.

Line 151: Please correct "One hundred fourth one".

Line 166, suggested format: "83.1% (95 % confidence intervals (CIs): 78.9%, 86.6%)".

Until line 177, you used "," as a separator between confidence intervals, but after line 177 "-" was used instead. Please keep the consistency.

Reviewers' comments:

Reviewer's Responses to Questions

**Comments to the Author**

1. If the authors have adequately addressed your comments raised in a previous round of review and you feel that this manuscript is now acceptable for publication, you may indicate that here to bypass the “Comments to the Author” section, enter your conflict of interest statement in the “Confidential to Editor” section, and submit your "Accept" recommendation.

Reviewer #1: All comments have been addressed

Reviewer #2: All comments have been addressed

Reviewer #3: All comments have been addressed

2. Is the manuscript technically sound, and do the data support the conclusions?

Reviewer #1: Yes

Reviewer #2: Yes

Reviewer #3: Yes

3. Has the statistical analysis been performed appropriately and rigorously? 

Reviewer #1: Yes

Reviewer #2: Yes

Reviewer #3: Yes

4. Have the authors made all data underlying the findings in their manuscript fully available?

Reviewer #1: Yes

Reviewer #2: No

Reviewer #3: Yes

5. Is the manuscript presented in an intelligible fashion and written in standard English?

Reviewer #1: Yes

Reviewer #2: Yes

Reviewer #3: Yes

6. Review Comments to the Author

Reviewer #1: (No Response)

Reviewer #2: I would like to thank the authors for addressing the issues. I have no further comments on the manuscript.

Reviewer #3: The authors addressed most of the previous comments. Some copy editing and other issues are still in the draft. I mention some of them here:

Please add the confidence interval (CI) measurement definition to the analysis section of the methods.

Some percentages were presented without decimals, some with one decimal, and some with two decimals. To maintain consistency, I suggest presenting all the percentages with one decimal.

Line 151: Please correct "One hundred fourth one".

Line 166, suggested format: "83.1% (95 % confidence intervals (CIs): 78.9%, 86.6%)".

Until line 177, you used "," as a separator between confidence intervals, but after line 177 "-" was used instead. Please keep the consistency.

7. PLOS authors have the option to publish the peer review history of their article (what does this mean?). If published, this will include your full peer review and any attached files.

Reviewer #1: **Yes: **Sina Azadnajafabad, MD, MPH

Reviewer #2: No

Reviewer #3: No

---

## [Author Response · Author response to Decision Letter 2]

10 Aug 2022

10/08/2022

PLOS ONE

Subject: Submission of revised manuscript entitled as “Advanced stage presentation and its determinant factors among colorectal cancer patients in Northwest Amhara Regional State Referral Hospitals of Ethiopia (PONE-D-22-07229R2).

Dear, Negar Rezaei, M.D., Ph.D, thank you for your email enclosing the Editorial member’s and the reviewer’s comments. We have carefully revised the manuscript and incorporated their comments accordingly. Our responses are given in point-by-point response below.

We hope the revised version is suitable for publication and look forward to hearing from you in due courses.

Sincerely

Mulugeta Wassie. 

Point by point responses to Editorial Board Member’s and Reviewers’ comments

Editor comments

Editor’s comment #01: Please consider the reviewer 3 comments 

Authors’ response: Thank very much for your suggestions. We have incorporated all the comments raised by reviewer 3 in the manuscript

Reviewers’ comments:

Reviewer #1: No comments.

Reviewer #2: No comments.

Reviewer #3: 

Please add the confidence interval (CI) measurement definition to the analysis section of the methods.

Authors’ response: Thank you very much for your constructive comment. We have added the confidence interval (CI) measurement definition in the manuscript on the analysis section.

Some percentages were presented without decimals, some with one decimal, and some with two decimals. To maintain consistency, I suggest presenting all the percentages with one decimal.

Authors’ response: Thank you very much for your constructive comments. We have tried to make consistent all decimals throughout the document using one decimal. 

Line 151: Please correct "One hundred fourth one".

Authors’ response: Thank you very much for your constructive comment. We have corrected it in the manuscript.

Line 166, suggested format: "83.1% (95 % confidence intervals (CIs): 78.9%, 86.6%)".

Authors’ response: Thank you very much for your constructive comment. We have corrected it as your suggestion.

Until line 177, you used "," as a separator between confidence intervals, but after line 177 "-" was used instead. Please keep the consistency.

Authors’ response: Thank you very much for your critical constructive comment. We have used "-" throughout the document to make consistency.

---

## [Editor Report · Decision Letter 3]

15 Aug 2022

Advanced stage presentation and its determinant factors among colorectal cancer patients in Amhara Regional State Referral Hospitals, Northwest Ethiopia

PONE-D-22-07229R3

Dear Dr. Wassie,

We’re pleased to inform you that your manuscript has been judged scientifically suitable for publication and will be formally accepted for publication once it meets all outstanding technical requirements.

Kind regards,

Negar Rezaei, M.D., Ph.D.,

Academic Editor

PLOS ONE
---

## [Editor Report · Acceptance letter]

24 Aug 2022

PONE-D-22-07229R3 

Advanced stage presentation and its determinant factors among colorectal cancer patients in Amhara Regional State Referral Hospitals, Northwest Ethiopia 

Dear Dr. Wassie:

I'm pleased to inform you that your manuscript has been deemed suitable for publication in PLOS ONE. Congratulations! Your manuscript is now with our production department. 

Kind regards, 

on behalf of

Dr. Negar Rezaei 

Academic Editor

PLOS ONE